

# Differences in lumbar spine intradiscal pressure between standing and sitting postures: a comprehensive literature review

Danuta Roman-Liu,  Joanna Kamińska and  Tomasz Tokarski

Ergonomics, Central Institute for Labour Protection–National Research Institute (CIOP-PIB), Warsaw, Poland

## ABSTRACT

**Background**. Musculoskeletal disorders (MSDs), especially in the lumbar spine, are a leading concern in occupational health. Work activities associated with excessive exposure are a source of risk for MSDs. The optimal design of workplaces requires changes in both sitting and standing postures. In order to secure such a design scientifically proved quantitative data are needed that would allow for the assessment of differences in spine load due to body posture and/or exerted force. Intradiscal pressure (IP) measurement in the lumbar spine is the most direct method of estimating spinal loads. Hence, this study aims at the quantitative evaluation of differences in lumbar spine load due to body posture and exerted forces, based on IP reported in publications obtained from a comprehensive review of the available literature.

**Methodology**. In order to collect data from studies measuring IP in the lumbar spine, three databases were searched. Studies with IP for living adults, measured in various sitting and standing postures, where one of these was standing upright, were included in the analysis. For data to be comparable between studies, the IP for each position was referenced to upright standing. Where different studies presented IP for the same postures, those relative IPs (rIP) were merged. Then, an analysis of the respective outcomes was conducted to find the possible relationship of IPs dependent on a specific posture.

**Results**. A preliminary analysis of the reviewed papers returned nine items fulfilling the inclusion and exclusion criteria. After merging relative IPs from different studies, rIP for 27 sitting and 26 standing postures was yielded. Some of the data were useful for deriving mathematical equations expressing rIP as a function of back flexion angle and exerted force in the form of a second degree polynomial equation for the standing and sitting positions. The equations showed that for the standing posture, the increase in IP with increasing back flexion angle is steeper when applying an external force than when maintaining body position only. In a sitting position with the back flexed at 20°, adding 10 kg to each hand increases the IP by about 50%. According to the equations developed, for back flexion angles less than 20°, the IP is greater in sitting than in standing. When the angle is greater than 20°, the IP in the sitting position is less than in the standing position at the same angle of back flexion.

**Conclusions**. Analysis of the data from the reviewed papers showed that: sitting without support increases IP by about 30% in relation to upright standing; a polynomial of the second degree defines changes in IP as a function of back flexion for for both postures.

Corresponding author
Danuta Roman-Liu, daliu@ciop.pl

There are differences in the pattern of changes in IP with a back flexion angle between sitting and standing postures, as back flexion in standing increases IP more than in sitting.

## INTRODUCTION

Musculoskeletal disorders (MSDs) are, and continue to be, a leading concern in occupational health. Symptoms of MSDs are often defined as pain in one or more regions of the body. MSDs may result in functional disability, low quality of life and socio-economic burden (*James et al., 2018*). Among others, non-specific low back pain is a main contributor to MSDs (*Maher, Underwood & Buchbinder, 2017*; *O'Keee, 2019*; *Oliveira et al., 2018*). Although the exact cause of low back pain remains less than fully defined, work-related physical factors were proven to play a substantial role (*Hartvigsen et al., 2018*).

Physical activities related to work tasks and associated with high exposure are a source of increased risk not only for MSDs (*Nordander et al., 2016*) but also for high blood pressure (*Clays et al., 2012*), cardiovascular disease (*Petersen et al., 2012*) and mortality (*Coenen et al., 2018*). The development of MSDs, however, is not solely associated with high physical performance work. Studies have confirmed that increasing sedentary behaviour at work is also considered to be an important risk factor for MSD development (*Nicoletti & Läubli, 2018*; *Bontrup et al., 2019*; *Waongenngarm, Areerak & Janwantanakul, 2018*). Epidemiological evidence arising from recent research suggests that prolonged sitting, accumulated throughout the entire day, is also associated with an increased risk of chronic diseases (type 2 diabetes, cardiovascular disease, certain types of cancer) (*Åsvold et al., 2017*; *Bailey et al., 2019*; *Lavie et al., 2019*), poor mental health (*Zhai, Zhang & Zhang, 2015*) and premature mortality (*Ekelund et al., 2019*). It should be remembered, however, that when a sitting position is replaced by a standing one, especially one which is consistently maintained for a long time, there is also an increased risk of developing MSDs (*Coenen et al., 2017*). Limiting sitting time by regularly interrupting prolonged periods of sitting by activities performed in standing postures, can alleviate musculoskeletal discomfort. This is due to the fact that exposure in daily work depends on the pattern of sequelae induced by work activities that are attributed to postures and exerted forces (*Roman-Liu, 2013*). Thus, work processes that allow shifts between standing and sitting change the pattern of physical activity, providing better muscle activation, and seem to be an optimal solution. Workplaces and work processes that allow or even impose on workers to alternate their posture between sitting and standing need a scientifically proven basis for them to be considered optimal solutions. This means that adequate load assessment procedures and quantitative data are needed that would allow for the assessment of differences in spine load due to body posture and/or exerted force (*Burdorf, 2010*; *Chiasson et al., 2012*). In this

context, knowledge of spine loads due to body posture and exerted forces is essential. The main measure of spinal loading is lumbar intradiscal pressure (IP).

IP measurement in the spine is the most direct method of estimating spinal loads. This has been extensively studied in the 1960s and the 1970s *in vivo* (*Nachemson & Morris, 1964*; *Sato, Kikuchi & Yonezawa, 1999*; *Takahashi et al., 2006*; *Wilke et al., 1999*) and *post mortem* (*McNally & Adams, 1992*; *Nachemson, 1960*; *Panjabi et al., 1988*). These pioneering studies presented measurements done in static conditions (*Nachemson, 1966*) and during movement (*Nachemson & Elfstrom, 1970*). Their results have created a reference point for assessing load in the human spine and are often cited by later works. At least two publications have been dedicated to a summary of the IP data reviewed in these studies. *Dreischarf et al. (2016)* presented a review of the nominal values of intradiscal pressure for standing, sitting and laying postures at the thoracic and lumbar parts of the spine. *Claus et al. (2008)* presented a review and comparison of various papers in regard to the sitting to standing ratio of intradiscal pressure. In both papers, the presented analysis referred to postures defined only as sitting or standing. Whereas an examination of the relationship between load and back pain in different sitting and standing body postures is meaningful in the context of the performance of work activities, solutions that would find the proper relationship of IP among different postures are dearly needed.

The measurement of IP was and still is challenging, as it is a highly invasive measurement method and is dependent on the measurement technique (sensors) and on the subjective characteristics of the person under study. The article by *Bashkuev et al. (2016)* suggests that a valid measurement of the nucleus pressure is a challenging task because the pressure sensor itself can influence the measured values. Large artefacts associated with the sensors were also reported in other studies (*McNally, Adams & Goodship, 1992*; *Nachemson, 1981*). The shape of the lumbar region (*Srbinoska et al., 2013*), as well as body mass and body height (*Hajihosseinali, Arjmand & Shirazi-Adl, 2015*; *Nachemson, 1966*) were also shown to impact the measurements. This means that in order to eradicate the errors related to those factors, relative values of IP are more reliable and recommended for application than absolute ones.

The aim of this study is, thus, to conduct a quantitative evaluation of load on the lumbar spine due to body posture and exerted forces, based on IP reported in publications indicated by a comprehensive review of the available literature. On this basis, the general research questions were formulated as follows: what are the quantitative differences in IP due to posture and exerted force; are there differences in IP between standing and sitting using the same back flexion; is there a significant difference in lumbar back IP between sitting upright and sitting in a relaxed posture? The study also examines whether there is any new research that would enhance our understanding of lumbar IP.

IP of the lumbar spine associated with body posture and exerted forces are the basis for direct assessment of workload for selected work activities and development of procedures for assessing exposure and risk of MSD not only for individual activities, but also for a set of activities that make up the work process.

The results of the literature review and analyses presented in the article are intended for researchers dealing with occupational safety, occupational medicine, designing
workstations and work processes, as well as designers of procedures for assessing and simulating spinal load in various working conditions.

The study consisted of two main parts. In the first part, a comprehensive review of the literature was performed to collect data from studies measuring IP. In the second part, an analysis of the respective outcomes directed at finding answers to the research questions was performed.

## COMPREHENSIVE REVIEW OF THE LITERATURE

In order to identify relevant studies of IP in sitting and standing postures, an electronic literature search of the following databases was undertaken: Medline (PubMed), ScienceDirect and PROQUEST. There were no limitations as to language or time of publication. Non-English language abstracts of potentially eligible published studies were translated for potential inclusion. The databases were searched in terms of title and abstract using the following keywords (sitting AND standing AND (force OR load OR pressure OR posture) AND (back OR spine)). There were no additional restrictions in ScienceDirect. In PubMed, the search was limited to case reports, classic articles, clinical studies, clinical trials and randomised controlled trials. The PROQUEST search included research journals, papers, case studies and full text peer reviews.

Inclusion and exclusion criteria covered participants and outcomes. Only studies reporting data on adult, living persons were included. In relation to the measures, studies were included if they presented measured intervertebral pressure. Only those studies that presented measures on various postures were included when one of those postures was the natural one (standing upright with arms down) and those where the IP referred to the lumbar part of the spine. Only sitting and standing postures formed the subjects of analysis.

The first step in the analysis was the removal of duplications. This was followed by the screening of titles and abstracts found in the comprehensive search in order to identify potential studies for analysis of full papers that satisfied the criteria for inclusion. Studies selected for further analysis were reported in a table according to the following extracted information: author, sample size, participant characteristics (age, body mass and body height) and posture.

The flow diagram for the review process is outlined in Fig. 1. Upon removing duplicates, the search returned a total of 501 results. The titles and abstracts of these results were screened, and 35 studies were found to be relevant to the problem of back load related to sitting and standing postures. The full text of each available article was assessed. Seven full text articles fulfilled the inclusion and exclusion criteria and were incorporated into the analysis. Two publications were added from the reference list of the analysed papers. Table 1 provides a list of all references, including summary information for each.

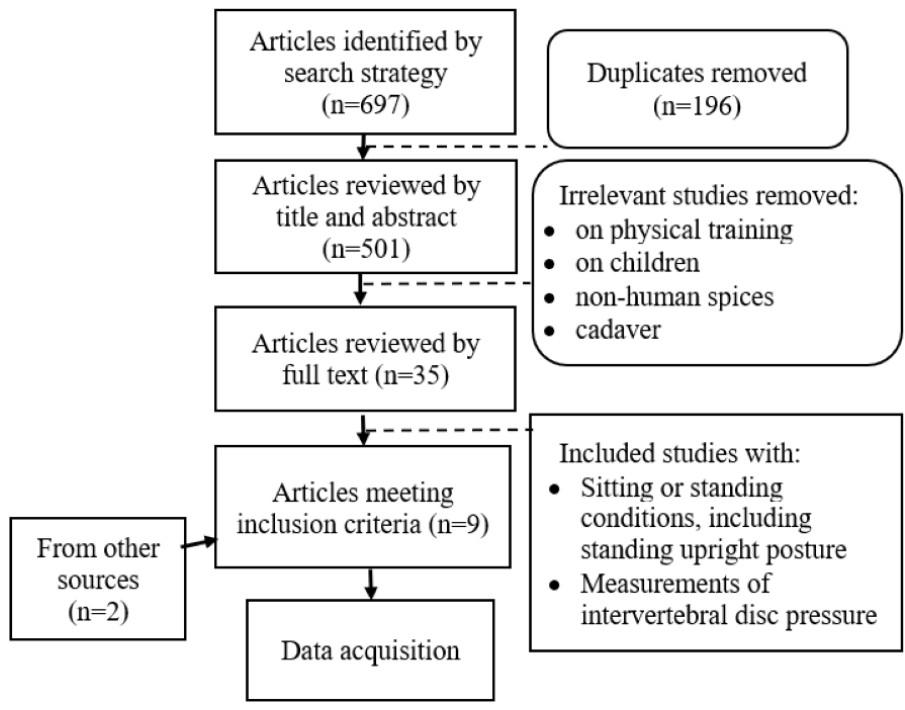

**Figure 1** Diagram of literature search.

## ANALYSIS OF OUTCOME MEASURES

### Data management

In order to compare the IP of the same posture among studies, unification of the study results was necessary. Data from each included study was entered onto a data extraction form, which assigned the value of the variable that measured IP to each individual body posture and study. It should be emphasized that body position is understood as the position and the value of external force. This means that the external force also differentiated the considered cases defined as body position. Then, for every posture within the study a relative intradiscal pressure (rIP) was calculated as IP for this posture, and juxtaposed to the IP for the standing upright posture. The rIP calculation process was dependent on the type of data provided by the article. In cases where a relative value of IP was provided, data were not further transposed, *e.g.*, this ratio was accepted for analysis as rIP. Some study provided absolute values of IP as means and standard deviations for groups of study participants or as one value for each participant and each posture. In those papers that provided IP for individual participants separately, the rIP for each participant was calculated first. According to this, the mean and standard deviation (SD) for this particular posture for all study participants was calculated. If the study presented absolute measures for defined postures as a mean and SD of a group of participants, the ratio of means of rIP was calculated with the application of the Ratio of Means (RoM) method, expressed in the

**Table 1  Summary of studies meeting the inclusion criteria and accepted for analysis (n, number of study participants).**

| Reference | n | Age | Body mass | Body height | |
|---|---|---|---|---|---|
| *Andersson, Ortengren & Nachemson (1977)* | 4 | 29 (26–34) | 52–69 | 169–177 | Standing: flexed 30° with load (0, 100, 300)N; standing with load 200 N and back flexion (0°, 10°, 20°, 30°, 40°, 50°) |
| *Nachemson (1966)* | 6 | – | 50–100 | – | Sitting: upright, 20° flexed, 20° flexed with 10 kg in each hand; standing: 20° flexed, 20° flexed with 10 kg in each hand |
| *Nachemson (1992)* (as cited by Wilke) | 1 | – | – | – | Standing flexed foreword; sitting actively straighten the back |
| *Nachemson & Morris (1964)* | 5 | 42 (9.19) | 78.4 (10.80) | 174.6 (9.55) | Sitting upright with (0; 4,5; 11,4) kg in each hand; sitting reclined on the table |
| *Nachemson & Elfstrom (1970)* | 6 | 23.3 (3.33) | 59.8 (3.31) | 170.8 (4.71) | Sitting: upright, flexed 20° with 10 kg in each hand; standing: upright with 10 kg in each hand, flexed 30° (0, 10) kg in each hand, bending sideways (0, 10) kg in each hand, twisting (0, 10) kg in each hand; lifting 20 kg (bending of back, bending of knees) |
| *Nachemson (1981)* | 1 | – | 70 | – | Sitting: upright, with lumbar support, back rest inclination 110°; standing: bend foreword (20°, 40°, 20° with 20 kg), flexed 20° and rotated 20° with 10 kg; lifting 10 kg (bending of back, bending of knees) |
| *Sato, Kikuchi & Yonezawa (1999)* | 28 | 45 (19–74) | 68 (45–88) | 1.65 (155–182) | Standing (flexion, extension); sitting (upright, flexion, extension) |
| *Wilke et al. (1999)*, *Wilke et al. (2001)* | 1 | 45 | 70 | 1.68 | Standing (flexed; holding 20 kg close to the body; holding 20 kg 60 cm away from the chest); sitting (relaxed without back-rest, actively straighten the back, with maximum flexion, with tight supporting elbows, slouched into the chair) |
| *Schultz et al. (1982)* | 4 | 21.8 (19–23) | 62.8 (56–76) | 174 (165–187) | Standing (holding 8 kg close to body, upright with arms extended, upright with arms extended holding 8 kg, flexed 30° with arms extended, flexed 30° with arms extended holding 8 kg); sitting (relaxed, in four different upper limb postures with and without 4 kg) |

following way (*Friedrich, Adhikari & Beyene, 2008*):

$$RL = (mean_{posture}/mean_{standing})$$

$$SD_{RL} = sqrt\{(1/n) * [(SD_{posture}/mean_{posture})^2 + (SD_{standing}/mean_{standing})^2]\}.$$

Expressing load in the lumbar spine as a relative value (rIP measure) makes results of different studies comparable regardless of the individual characteristics of the group of study participants or the measurement technique. If more than one study presented data for the same posture, the final step included grouping data of rIP from various studies together and presenting it as one representative measure. In order to do so, weighted means

and pooled SD was calculated, as proposed by *Killeen (2005)*:

$$rIP_{\text{pooled}} = (n_1 \cdot rIP_1 + \ldots\ldots + n_k \cdot rIP_k)/(n_1 + \ldots n_k)$$

$$SD_{\text{pooled}} = \text{sqrt}[\{(n_1 - 1) \cdot SD_1^2 + \ldots.. + (n_k - 1) \cdot SD_k^2\}/(n_1 + n_2 + \ldots\ldots + n_k - -k)]$$

where n is the number of participants in a given study.

Intradiscal pressure was measured in nine studies with altogether 37 sitting postures and 46 standing ones (excluding standing upright). There were postures, for which IP was presented in a number of studies, while others in only one. Postures have been described in various ways in different studies. Some descriptions were quantitative by presenting the values of the angles, while others were qualitative presenting a qualitative description in terms of "flexion", "bending", "twisting" or "extension". This means that, in some cases, details that would allow for a clear definition of the posture were not presented. In those cases where postures were defined quantitatively and there was enough data, mathematical equations that expressed rIP as a function of angles or external force were developed.

Models of changes in rIP were developed using regression analysis of the Statistica 10 package. As a model of rIP changes, a formula was adopted that expressed a function of two variables (back flexion angle and exerted force) and ensured the best fit of the model calculations to the experimental data. The fit was tested with the value of the Spearman's correlation coefficient.

## Data obtained from reviewed studies

Most of the studies presented absolute values of IP for each person and each posture (*Andersson, Ortengren & Nachemson, 1977*; *Nachemson, 1966*; *Nachemson & Morris, 1964*; *Nachemson & Elfstrom, 1970*; *Sato, Kikuchi & Yonezawa, 1999*). In those cases, for each person and each posture, rIP was calculated first followed by the mean and standard deviation for a group of this study participants. *Schultz et al. (1982)* performed a study on four volunteers and presented mean values of IP for each posture. The study by *Wilke et al. (1999)* and *Wilke et al. (2001)* presented IP for one person as both absolute and relative values (in relation to standing upright). *Wilke et al. (1999)* and *Wilke et al.*'s (*2001*) results concerned the same single participant. As some postures differ between these articles, both were included. *Wilke et al. (1999)* and *Wilke et al. (2001)* measured intradiscal pressure over a broad range of postures and activities. Tables 2 and 3 report the IP of lumbar back, as calculated in the reviewed papers, for both standing and sitting postures respectively.

In general terms, IP in the lumbar spine is higher in a sitting posture that in upright standing. There are cases, however, when IP is lower. Slouching on a chair, relaxing and reclining on a table are postures that induce lower IP than standing upright by 50–60% (Table 3). Also, sitting bent foreword in a 40° angle with the elbows resting on the thighs lowers lumbar back IP by 14% compared to standing upright. There are observable differences in IP between sitting on a chair and on a ball. When sitting on a ball with a straight back, IP is very similar to standing upright. When sitting on a ball in a flexed posture, IP increases by 30% (Table 3). The *Wilke et al. (1999)* and *Wilke et al. (2001)* study reports an increase in IP when sitting flexed with the head in front of the body. As shown in Table 3, this posture increases IP by about 80% compared to sitting upright.

**Table 2  Values of relative intradiscal pressure (IP) at lumbar back for different standing postures presented in the reviewed studies.**

| Reference | Conditions | Mean | sd |
|---|---|---|---|
| | *standing in different postures without external force* | | |
| *Andersson, Ortengren & Nachemson (1977)* | flexed 30° | 1.60 | 0.500 |
| *Nachemson & Elfstrom (1970)* | bending forward 30° | 2.09 | 0.076 |
| | bending sideways | 1.22 | 0.142 |
| | twisting | 1.12 | 0.000 |
| *Nachemson (1966)* | leaning foreword 20° | 1.50 | 0.019 |
| *Nachemson (1981)* | Forward bend 20° | 1.2 | |
| | Forward bend 40° | 2 | |
| *Nachemson, (1992)* | bent foreword | 1.50 | |
| *Sato, Kikuchi & Yonezawa (1999)* | flexion | 2.67 | 0.728 |
| | extension | 1.20 | 0.447 |
| *Wilke et al. (1999), Wilke et al. (2001)* | bent foreword 40° | 2.20 | |
| | extension 19° | 1.2 | |
| | extension 10° | 1.1 | |
| | flexion 10° | 1.2 | |
| | flexion 20° | 1.6 | |
| | flexion 30° | 1.9 | |
| | flexion 36° | 2.16 | |
| *Schultz et al. (1982)* | upright with arms extended | 1.11 | |
| | flexed 30° with arms exteded | 3.85 | |
| | *standing in different postures with external force* | | |
| *Andersson, Ortengren & Nachemson (1977)* | back flexion 30° with load 50 N in each hand | 2.40 | 0.500 |
| | back flexion 30° with load 150 N in each hand | 3.60 | 0.500 |
| | back flexion 10° with load 100 N in each hand | 1.60 | 1.000 |
| | back flexion 20° with load 100 N in each hand | 2.40 | 0.500 |
| | back flexion 40° with load 100 N in each hand | 3.60 | 0.500 |
| | back flexion 50° with load 100 N in each hand | 4.40 | 1.000 |
| | back flexion 30° with load 100 N in each hand | 3.00 | 0.500 |
| *Nachemson & Elfstrom (1970)* | bending sideways with 10 kg in each hand | 1.94 | 0.364 |
| | twisting with 10 kg in each hand | 1.79 | 0.172 |
| | lifting of 20 kg with bending of back | 3.81 | 0.524 |
| | upright with 10 kg in each hand | 1.52 | 0.277 |
| | bending 30° with 10 kg in each hand | 2.78 | 0.253 |
| *Nachemson (1981)* | Forward bend 20° with 20 kg | 2.4 | |
| | Forward bend 20° and rotated 20° with 10 kg | 4.2 | |
| | *standing with external force* | | |
| *Nachemson & Elfstrom (1970)* | lifting of 20 kg with bending of knees | 2.47 | 0.494 |
| *Nachemson (1966)* | foreword leaning 20° with 10 kg in each hand | 2.19 | 0.096 |

**Table 2** (*continued*)

| Reference | Conditions | Mean | sd |
|---|---|---|---|
| *Nachemson (1981)* | lifting of 10 kg with bending of knees | 3.4 | |
| | lifting of 10 kg with bending of back | 3.8 | |
| | holding 5 kg with arms extended | 3.8 | |
| *Wilke et al. (1999)* | holding 20 kg close to the body | 2.20 | |
| | lifting of 20 kg with bending of back | 4.60 | |
| | lifting of 20 kg with bending of knees | 3.40 | |
| | holding 20 kg, 60 cm away from the chest | 3.60 | |
| *Schultz et al. (1982)* | upright with arms close to the chest and holding 8 kg | 2.04 | |
| | upright with arms foreword and holding 8 kg | 2.48 | |
| | flexed 30° with arms foreword and holding 8 kg | 6.00 | |

### Relationship of IPs dependent on a specific posture

Relative IP values (rIP) obtained for each of postures that are presented in Tables 2 and 3 were grouped when referring to the same posture. After pooling rIP values from different studies for the same posture, rIP was presented for 27 different sitting postures and 26 standing ones. For each posture, the aggregated relative intradiscal pressure was calculated. Some of the results of these calculations were used in order to determine a mathematical equation that expresses pooled rIP as a function of back flexion angle and exerted force.

In the case of standing posture for exerted force close to 200 N (10 kg in each hand), experimental data were obtained for six back flexion angles. Data were also available for seven back postures without external load, and for 30° back flexion with 100 N and 300 N. This determined a second degree polynomial equation of rIP as a function of back flexion angle and exerted force (Eq. 1).

Relative IP values allowed to develop an equation expressing rIP as a function of back angle flexion and exerted force for the sitting posture as well (Eq. 2). For the sitting posture, experimental data were available for four back positions without external force, 0° back flexion with 90 N and 230 N force exerted, and 20° back flexion with 200 N force. Due to the rough definition of postures, in the first step data that referred to the upright sitting posture were analysed separately from those that referred to a relaxed/slumped sitting. Data for those two cases showed lack of differences in rIP values, which proved that both of those postures can be treated as one and data can be pooled.

$$rIP_{standing} = 1.08 + 0.005 * A + 0.0012 * F + 0.0006 * A^2 + 0.0001 * A * F \quad (1)$$

$$rIP_{sittin} = 1.35 + 0.007 * A + 0.0032 * F + 0.0001 * A^2 + 0.0001 * A * F \quad (2)$$

where

A–back flexion angle (°)

F–exerted force (N)

Figure 2 presents the experimental data (mean and sd) and the rIP values calculated with the Eq. (1) for standing posture. Changes in rIP with back flexion angle for 0N and for 200 N can be approximated by second degree curves with increase for 200 N steeper than when

**Table 3** Values of relative intradiscal pressure (IP) at lumbar back for different sitting postures presented in the reviewed studies.

| Reference | Conditions | Mean | sd |
| --- | --- | --- | --- |
| | sitting upright or relaxed without external force | | |
| Nachemson & Elfstrom (1970) | without support | 1.36 | 0.222 |
| Nachemson & Morris (1964) | upright | 1.43 | 0.239 |
| Nachemson, as cited by Wilkie | actively straighten the back | 1.45 | |
| Nachemson (1966) | upright with arms and back unsupported | 1.43 | 0.022 |
| Nachemson (1981) | Upright sitting, without support | 1.4 | |
| | Sitting with lumbar support at inclination 110° | 0.8 | |
| Sato, Kikuchi & Yonezawa (1999) | upright | 1.24 | 0.330 |
| Schultz et al. (1982) | relaxed | 1.18 | |
| Wilke et al. (1999), Wilke et al. (2001) | actively straighten the back | 1.10 | |
| | relaxed without backrest | 0.92 | |
| | sitting on a ball with straight back | 1 | |
| Schultz et al. (1982) | relaxed | 1.8 | |
| | sitting in various positions without external force | | |
| Nachemson (1966) | foreword leaning 20° | 1.93 | 0.003 |
| Nachemson & Morris (1964) | reclined on the table | 0.58 | 0.201 |
| Sato, Kikuchi & Yonezawa (1999) | flexion | 2.26 | 0.553 |
| | extension | 1.52 | 0.606 |
| Wilke et al. (1999) | with maximum flexion | 1.66 | |
| | bent foreword 40° with tight supporting elbows | 0.86 | |
| | slouched into the chair | 0.54 | |
| | bending foreword 20° | 1.26 | |
| | bending foreword 40° | 1.66 | |
| | flexed with head in front of the body | 1.8 | |
| | on a ball in flexed posture | 1.3 | |
| | sitting with external force | | |
| Nachemson (1966) | foreword leaning 20° with 10 kg in each hand | 2.73 | 0.185 |
| Nachemson & Elfstrom (1970) | leaning forward 20°, 10 kg in each hand | 2.73 | 0.309 |
| Nachemson & Morris (1964) | upright with 4.5 kg in each hand | 1.65 | 0.238 |
| | upright with 11.4 kg in each hand | 2.12 | 0.291 |
| | sitting upright with different positions of upper limb | | |
| Schultz et al. (1982) | upper limb in position A | 1.15 | |
| | upper limb in position A holding 4 kg | 2.11 | |
| | upper limb in position B | 1.11 | |
| | upper limb in position B holding 4kg | 1.78 | |
| | upper limb in position C | 1.15 | |
| | upper limb in position C holding 4kg | 2.22 | |
| | upper limb in position G | 1.11 | |
| | upper limb in position G holding 4kg | 2.00 | |
| | upper limb in position I | 1.04 | |
| | upper limb in position I holding 4kg | 1.93 | |
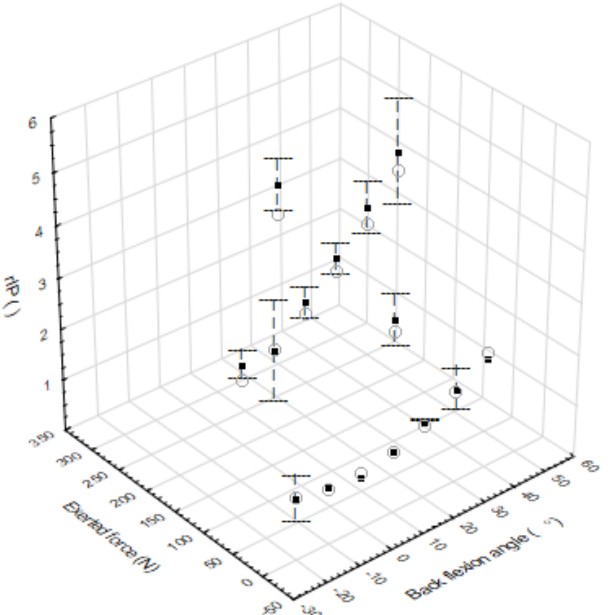

■ rIP measured (mean)
---- sd of mean rIP
○ rIP calculated

**Figure 2** Values of relative intradiscal pressure (rIP) at the lumbar spine as a function of angle of back flexion angle and external force in the standing posture.

the external force equals to zero. rIP calculated with the equation showed high Spearman correlation coefficient (equal to 0.974) with experimental data. An extension posture, compared to standing upright, does not significantly change lumbar back load. *Nachemson & Elfstrom (1970)* calculated rIP for flexed postures, without providing the value of angle flexion, as equal to 2.54 (0.73). According to the calculations with the equation, this would suggest back flexion of about 45°.

Figure 3 presents rIP at the lumbar spine for different angles of back flexion and different values of loads carried in the hands for sitting posture. The experimental data showed high Spearman correlation coefficient (equal to 0.964) with calculations with the model. Changes due to back flexion are smaller than those obtained due to standing. Figure 3 presents data for back flexion with 20° and external force of 0 N and 200 N. Adding this external force, increased rIP by about 50%, which is a similar increase to adding 200 N to the standing posture flexed by 20°. The figure shows IP as a linear function of external force.

In order to express differences in IP between sitting and standing postures, the ratio of sitting to standing for the same angle of back flexion was calculated. Figure 4 presents the values of this ratio and illustrates that for back flexion angles equal to 0° and 10°, IP is higher for sitting than for standing. When the angle is 20°, there are no differences in IP,

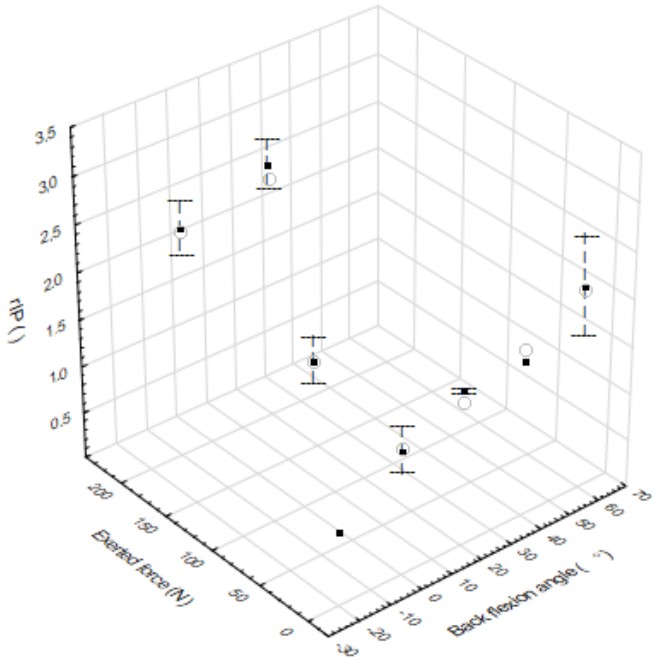

- ■ rIP measured (mean)
- ---- sd of mean rIP
- ▫ rIP calculated

**Figure 3** **Relative intradiscal pressure at lumbar back as a function of back flexion angle and external force in the sitting posture.** Also presented is the value of rIP for back extension with lumbar support that equals to 0.8.

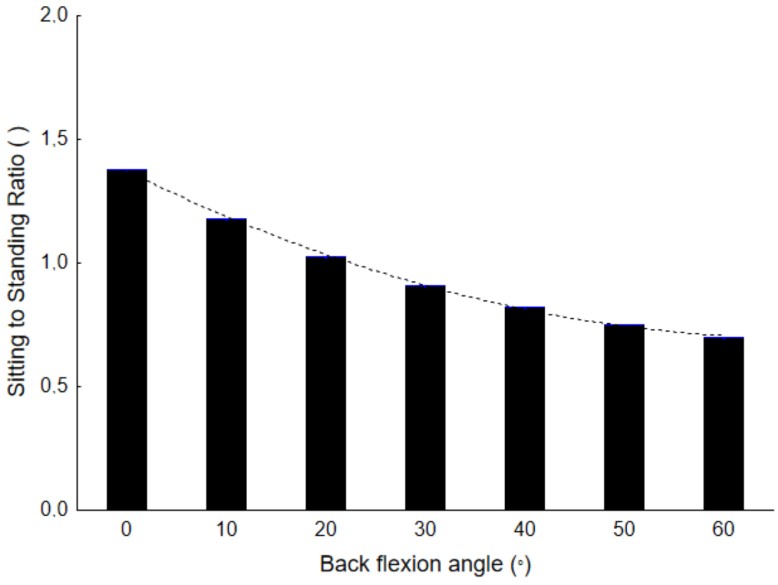

**Figure 4** **Ratio of sitting to standing posture for back flexion of the same angle.**

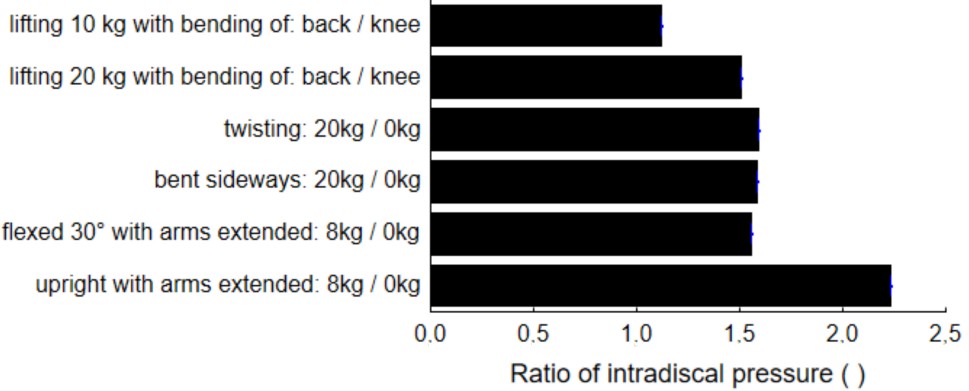

**Figure 5** **Difference in intradiscal pressure due to the addition of external force or due to a different technique in lifting the load.**

while for bigger angles, IP in sitting is lower than in standing for the same angle of back flexion.

Figure 5 presents differences in IP between cases with and without external force for standing postures, as well as a quantitative increase in IP due to differences in posture when lifting a load. Adding 8 kg of force to standing in 30° flexion increases the lumbar load by about 50%. The same effect is achieved by adding 10 kg to each hand in a twisting or bending posture. The figure also shows about 50% increase in lumbar back IP when a load of 20 kg is lifted with back flexion in comparison to when the load is lifted while bending the knees. Lifting 10 kg gives only about 10% increase in IP. The highest increase in IP was observed when a load of 8 kg was added to the upright standing posture with arms extended. The arms extended posture, however, has not been defined more precisely.

*Schultz et al. (1982)* measured IP in the sitting posture for four different upper limb positions. Position A required the adduction of an extended upper limb by moving in front of the body by about 75°. In positions B and C, the angle of adduction was 45°. Position B was closer to the body. In position C, the upper limb was fully extended. For each position, IP was measured in two variants, without external force and with 4 kg at hand. In order to express quantitatively the increase of IP under the influence of external force, for each position, a comparison of the ratio of IP with external force to IP without external force was drawn (Fig. 6). In most postures, adding 4 kg nearly doubles IP measured for a posture without external force. It can be noted that addition of external force in posture B (closer to the body) increases IP at the lumbar spine less significantly than in other postures.

## DISCUSSION

In order to examine back load due to posture, this study analysed relative lumbar back load, expressed as the ratio between IP in any posture and IP in an upright standing position. The analysis showed differences not only between sitting and standing postures but also among the variety of sitting and standing positions. It is commonly assumed that sitting relative to standing causes higher intradiscal pressure in the lumbar spine, alongside

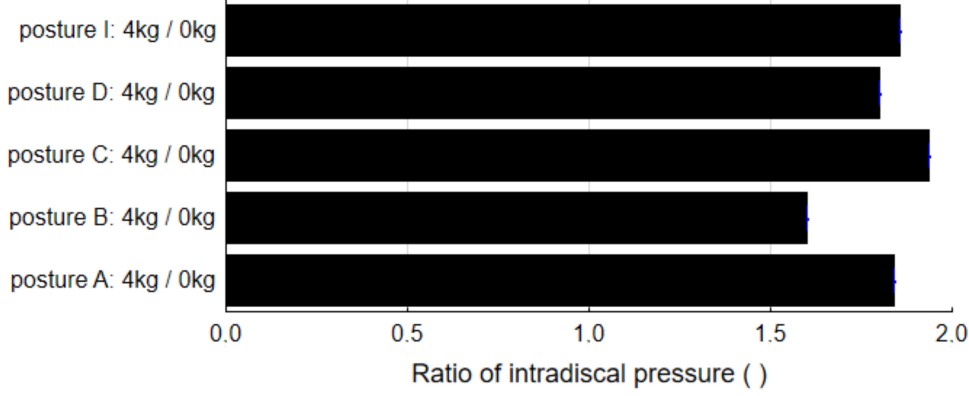

**Figure 6 Differences in intradiscal pressure due to the addition of external force to different postures of an upper limb in the sitting upright back posture.**

lower back pain, and disc degeneration or even rupture (*Castanharo, Duarte & McGill, 2014*; *Pynt, Mackey & Higgs, 2008*). The analysis performed in this study generally supports that statement. The calculation of relative IP, its integration and the comparative analysis performed in this study showed that, overall, sitting postures are more strenuous than standing ones. Intradiscal pressure in upright sitting related to upright standing was in the range of 1.24 (*Sato, Kikuchi & Yonezawa, 1999*) to 1.45 (*Nachemson, 1992*, as cited by Wilke, 1999). Aggregated relative IP for upright sitting equalled to 1.34, while for relaxed sitting this was 1.26. This means that both unsupported sitting, and sitting described as relaxed increase the load by about 30% in relation to standing upright. Higher load during sitting is also argued in studies performed with other measurement techniques. *Quinnell, Stockdale & Willis (1983)* used the equilibrium pressure technique to show that IP in upright sitting increases by 30% compared with upright standing. Similarly, *Huang et al. (2016)* used the motion capture approach and musculoskeletal modelling to show an increase in pressure of 39% in L3/L4, which doubled in L4/L5. Conversely, other studies suggest that compression forces and flexion bending moments were not higher for sitting than for standing. *Rohlman et al.'s (2001)* patient studies showed that the ratio comparing slumped sitting to upright standing was 0.85. Also, *Leivseth & Drerup (1977)* showed that standing compered to sitting causes grater shrinkage of the lumbar spine, which would suggest that standing causes higher spinal load. These differences in study results may be due to the fact that postures are poorly reproducible (*Schmidt et al., 2018*; *Marks et al., 2003*). The spine can adopt more than one posture in sitting. For example, sitting upright uses a lordotic posture, whereas slump sitting a kyphotic one (*Chen et al., 2019*). When sitting is defined as upright differences in spinal shape may also occur. Thus, in order to secure reproducibility, a clear definition of the posture must be provided alongside also the proper spinal shape. The studies analysed in this article, in most cases, provided a rough definition of posture, while they did not monitor spinal shape. The analysis performed in this article distinguished between cases of upright sitting and sitting described as slumped or relaxed. A comparison of pooled relative IP for those two cases, however, did not present

any differences. It is likely that variability in spinal shape and IP are caused by differences in the manner in which individuals sit (*Srbinoska et al., 2013*). *Castanharo, Duarte & McGill (2014)* showed that during sitting, movement may change from a slouched position to an upright trunk position using two patterns (lumbopelvic and thoracic). The lumbopelvic strategy positions the lumbar spine closest to the neutral posture. Following this, it is recommended that one's sitting posture should have a lordotic lumbar spinal curve similar to that achieved when standing (*Pope, Goh & Magnusson, 2002*; *Castanharo, Duarte & McGill, 2014*).

The analysed studies provided data that allowed for a quantitative expression of relative IP in the lumbar spine as a polynomial function of back flexion angle and exerted force. Such a relationship was determined for both sitting and standing postures. The developed relationships provide a tool that can be useful for optimising working conditions. The analysis of the available studies showed that if the back flexion angle is larger than 20°, IP for standing is higher than in sitting with the same back flexion angle. This supports the results of epidemiology studies that provided conflicting evidence as to whether sitting with a flexed spine was worse for spinal health and back pain than standing (*Battie et al., 1995*). It has also been argued that IP of the lumbar spine depends strongly on the back flexion angle and that back extension does not increase IP. The analysis clearly proves the positive impact of elbow support, showing that elbows resting on thighs decreases the IP of the lumbar spine by about 50% compared to sitting in the same flexion without elbow support.

The present literature review highlighted the lack of updated studies on healthy subjects presenting intradiscal pressure as dependent on various postures. Similarly, other recent literature reviews on IP showed the near total lack of such studies in the twenty-first century (*Dreischarf et al., 2016*; *Claus et al., 2008*). IP measurement is the most direct measurement technique providing data on load in the lumbar spine. Taking into account, however, that IP measurements are challenging, as they are highly invasive, other techniques to estimate spinal loads *in vivo* conditions are used. *Quinnell, Stockdale & Willis (1983)* measured equilibrium pressure for radiographically normal discs in seven patients and presented absolute values for each posture as a mean and standard deviation. *Leivseth & Drerup (1977)* compared standing to sitting based on the measurement of shrinkage of the lumbar spine. *Huang et al. (2016)* assessed lumbar back load with models and motion capture. Also, abdominal pressure has been presented as a measure of back load (*Andersson, Ortengren & Nachemson, 1977*). Model calculations are also common (*Zanjani-Pour et al., 2016*).

Trunk muscles play an important role in maintaining spinal stability postures (*Cholewicki & Van Vliet 4th, 2002*) and, thus, surface electromyography (EMG) of lumbar erector spinae muscles could have emerged as an indispensable complementary tool for the non-invasive estimation of spinal loads under various conditions (*Freivalds et al., 1984*; *Jørgensen et al., 1985*). Studies, however, have proven that higher muscle tension does not always reflect higher spinal load (*Reeve & Dilley, 2009*). The results of *Schultz et al. (1982)* showed that EMG amplitude can be twice as high in standing for a spine flexed by 20° or 40° compared to standing with the spine fully flexed. Discrepancies can be due to the fact that measurements taken with surface electrodes depend on many factors, one of which

is muscle length. There is evidence that the surface electromyography variables depend on the length of lumbar erector spinae muscles due to their complex anatomy (*Petrofsky et al., 1982*). Flexion/extension movement of the back is linked to changes in the distance between the origins and insertions of lumbar erector spinae muscles and, consequently, the length of the muscle, which can bias the results (*Noguchi et al., 2019*). Also, such factors as the proportion of muscle fibres (*Gerdle et al., 1997*; *Kupa et al., 1995*) and unwanted signals from surrounding muscles, *e.g.*, crosstalk (*Lowery, Stoykov & Kuiken, 2003*), play a role in the analysis and assessment of muscle contraction. This means that these methods, as an alternative to IP measurement, should be applied with caution.

MSDs are related to the duration, frequency and magnitude of exposure at the work stand (*Hembecker et al., 2017*). Exposure is a function of variables that describe posture, force and time sequences (*Roman-Liu, 2013*). In this context, knowledge of spinal loads due to body posture is essential for the design of adequate back load assessment procedures. This renders the relationship between back load and different body postures as extremely important. This study presents this relationship in its widest possible extent based on available data. As a result of the analysis carried out in this article, the load on the spine for various positions was presented in the form of relative IP values. Taking into account the differences between the anthropometric measures of different people in terms of disc diameter and health state, it seems that relative measures are the most appropriate method for differentiating back load between postures in the assessment of work-related exposure.

## STRENGTHS AND LIMITATIONS

The main subject of the presented article, which is intradiscal pressure measurements, can be treated as both a strength and a limitation of the analysis presented in this study. In order to determine load on the spine, the most reliable studies use the direct measurement of IP. Since IP measurement, however, is a highly invasive measurement technique, it obviously reduces the number of available data. Thus, the small number of studies that were included in this analysis and the relatively small number of participants in each study is a major limitation of the present study. At the same time, the strength of this article lies on the in-depth, comprehensive analysis of the available direct measurements of IP results.

Another serious limitation that impedes this study's objectives was, in many cases, the lack of quantification of body posture. Some studies described the posture as bent, flexed or twisted without specifying the angle value. This limited the number of equations developed that could express IP as a function of angles or forces. A further limitation refers to the definition of the sitting posture. In some studies, the sitting position was not clearly defined. For the sake of analysis, it was simply assumed that the posture was upright sitting, even if differences may be equally noted in upright sitting positions, as shown in this article.

It has also to be pointed out as a limitation that no newly published studies were found that would enhance the knowledge about IP of the lumbar spine. The inclusion of more recent studies would make the results of this study more representative in terms of the lifestyle nowadays, thus increasing the possibility of generalization of the study's conclusions. A review of the literature did not yield studies conducted after 2001. The lack

of such studies in the last 20 years may be due to the invasiveness of direct measurement of intradiscal pressure and current ethical standards that prevent such measurements in healthy people. As a result the IP measurements cited in this article may be the only ones available.

## CONCLUSIONS

The analysis of the data allowed for the development of mathematical equations that express IP as a function of back flexion angle and exerted force for sitting and standing postures. Based on these equations, it can be stated that a polynomial of second degree with two variables defines changes in IP as a function of back flexion. It also shows that external force increases IP with an angle of back flexion more than if force was not applied.

Differences in changes in IP between sitting and standing postures with an angle of back flexion have also been demonstrated. In back flexion lower than 20°, IP in sitting is higher than in standing postures. When the flexion angle is larger, IP in standing postures is higher than in sitting. The present study confirmed that upright sitting and sitting described as relaxed increase the load by about 30% in relation to standing upright.

It should be emphasized, however, that the results of the analysis including the developed equations should be treated with caution. The small number of studies that were included in this analysis and the relatively small number of participants in each study may raise concerns about the strength of the conclusions.

### Funding

This article is based on the results of a research task carried out within the scope of the sixth stage of the National Programme "Governmental Programme for Improvement of Safety and Working Conditions" supported within the scope of state services by the Ministry of Family and Social Policy. It was published in terms of the task no. 3.ZS.11 entitled "Determination of biomechanical and mental load as risk components for the development of musculoskeletal disorders". The Central Institute for Labour Protection – National Research Institute is the Programme's main co-ordinator. There was no additional external funding received for this study. The funders had no role in study design, data collection and analysis, decision to publish, or preparation of the manuscript.

### Grant Disclosures

The following grant information was disclosed by the authors:
The sixth stage of the National Programme "Governmental Programme for Improvement of Safety and Working Conditions" supported within the scope of state services by the Ministry of Family and Social Policy.
The Central Institute for Labour Protection–National Research Institute is the Programme's main co-ordinator.

## Competing Interests

The authors declare there are no competing interests.

## Author Contributions

- Danuta Roman-Liu conceived and designed the experiments, analyzed the data, prepared figures and/or tables, authored or reviewed drafts of the article, and approved the final draft.
- Joanna Kamińska performed the experiments, analyzed the data, prepared figures and/or tables, and approved the final draft.
- Tomasz Tokarski performed the experiments, analyzed the data, prepared figures and/or tables, and approved the final draft.

## Data Availability

The article presents all data obtained from reviewed studies.

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
