# Peer review of "Differences in lumbar spine intradiscal pressure between standing and sitting postures: a comprehensive literature review"

_PeerJ, doi:10.7717/peerj.16176_

## Round 0.1 · original submission · Major Revisions

There are major revisions needed for this manuscript.

**Language Note:** PeerJ staff have identified that the English language needs to be improved. When you prepare your next revision, please either (i) have a colleague who is proficient in English and familiar with the subject matter review your manuscript, or (ii) contact a professional editing service to review your manuscript. PeerJ can provide language editing services - you can contact us at copyediting@peerj.com for pricing (be sure to provide your manuscript number and title). – PeerJ Staff

Reviewer 1 ·

Basic reporting

- The authors proposed to use data from previous publications to conduct a quantitative evaluation of load on the lumbar spine due to body posture and exerted forces. Relative intradiscal pressure (rIP) value was used in the analyses. rIP is defined as the ratio between IP in any posture and IP in an upright standing position. But in Figure 5 (x-axis), Relative IP values were compared as differences in IP between cases with and without external force. The inconsistencies definition on rIP is a bit confusing and it will be helpful to explicitly specify the reference level and stick to it consistently throughout the study to ensure the validity and reliability of the rIP values reported.

- In line 124, replace “mas” with “mass”.

Experimental design

- In the data management section, since Relative intradiscal pressure (rIP) value was used for analysis, it will be important to make sure the baseline IP values are comparable across different studies. Given the wide range of publication years of previous studies (1966-2001), how is that guaranteed?

- In the IP model section (lines 258-274), the authors opted to model relative IP as a function of back flexion angle for various external forces (ranging from 0N to 200N). Instead of creating separate models for each external force, a more comprehensive approach could have been employed by including external forces as predictive variables in the formula. This inclusion would have allowed for a better understanding of the relationship between rIP, external forces, and back flexion angle within a single unified model.

- The authors put forth a second-degree polynomial equation to represent relative IP as a function of back flexion angle. To enhance the clarity of their methodology, it would be beneficial to elucidate the process by which the degree of the polynomial function was selected and how well it fits the data. This explanation would provide valuable insights into the appropriateness of their modeling approach.

Validity of the findings

- Data from various historical publications were used. A concern that arises from this approach is the overall small total sample size (56 subjects in total). The limited sample size poses a challenge in drawing robust conclusions and generalizations. Although the authors mention this as a limitation in the discussion section, the authors need to explicitly communicate this concern in the results and conclusion sections to ensure a comprehensive understanding of the study's findings. Readers need to be made aware of the potential implications on the strength of the conclusions drawn from the data.

- The current literature review exclusively encompasses studies conducted before 2001. To enhance the overall applicability and relevance of the results to contemporary lifestyles, it would be beneficial to incorporate studies conducted within the past 20 years. By including more recent research in this area, the findings are likely to be more representative of current people's lifestyles, thus increasing the generalizability of the study's conclusions.

·

Basic reporting

Good clear English
Extensively covered the existing literature

Experimental design

Methods detailed out well

Validity of the findings

Sound conclusion.

---

## Round 0.2 · accepted · Accept

Dear Authors, Thank you for your revised article admission which has been accepted.

Reviewer 1 ·

Basic reporting

I think that the authors have adequately addressed the comments made by the reviewers in the revised version of the manuscript. Therefore, I have no further comments.

Experimental design

I think that the authors have adequately addressed the comments made by the reviewers in the revised version of the manuscript. Therefore, I have no further comments.

Validity of the findings

I think that the authors have adequately addressed the comments made by the reviewers in the revised version of the manuscript. Therefore, I have no further comments.